# The Use of [^11^C]C-Methionine in Diagnostics of Endocrine Disorders with Focus on Pituitary and Parathyroid Glands

**DOI:** 10.3390/ph18020229

**Published:** 2025-02-07

**Authors:** Adam Daniel Durma, Marek Saracyn, Maciej Kołodziej, Katarzyna Jóźwik-Plebanek, Grzegorz Kamiński

**Affiliations:** Department of Endocrinology and Radioisotope Therapy, Military Institute of Medicine—National Research Institute, Szaserów 128, 04-141 Warsaw, Poland

**Keywords:** methionine, 11C, [^11^C]C-methionine, [^11^C]C-MET, PET/CT, PET/MRI, pituitary, adenoma, hyperparathyroidism, primary hyperparathyroidism, tertiary hyperparathyroidism, HPT, parathyroid adenoma, PA

## Abstract

The rapid development of nuclear medicine offers vast opportunities for diagnosing neoplasms, particularly in endocrinology. The use of the [^11^C]C-methionine radiotracer is currently limited due to its physical properties and the complex production process. However, studies conducted so far have demonstrated its utility in PET imaging, helping to detect lesions that often remain elusive with other modalities. This systematic review focuses on [^11^C]C-methionine in diagnosing hyperparathyroidism and pituitary tumors, highlighting its high effectiveness, which can be crucial in diagnosis. Despite some disadvantages, it should be considered when available, especially when other modalities or radiotracers fail.

## 1. Introduction

Nuclear medicine is one of the fastest-growing fields in recent years, both in terms of the availability of new therapeutic and diagnostic methods. Positron emission tomography (PET) using short-lived isotopes that undergo beta plus (β+) decay, i.e., emitting positrons, enables non-invasive localization of disease lesions with a specific type of metabolism or the presence of expression of specific proteins and/or receptors.

The principle of the PET diagnostic method is based on β+ decay in the atomic nucleus, where a proton is transformed into a neutron. This process is accompanied by the emission of a positively charged positron characteristic for each isotope [1]. After traveling a certain distance and losing some energy, the positron collides with an electron, and annihilation of those particles occurs, resulting in the emission of two quanta of γ radiation. These quanta are detected, and activity distribution is computer processed to provide diagnostic images [2].

In order to improve the image quality and to project the foci of high radiotracer accumulation onto anatomical structures, computed tomography (CT) is performed simultaneously with PET acquisition (usually as a low-dose examination) without the use of intravenous contrast. Such an examination allows for obtaining a combined image, the so-called PET/CT fusion [3]. Less frequently, functional PET images are combined with anatomical magnetic resonance images (MRI) [4,5,6].

Due to this process, PET acquisition combined with the aforementioned low-dose CT allows for a decrease in radiation dose [7,8].

Globally, the isotope most commonly used in PET is fluorine-18 (^18^F) in a form of ^18^F-labeled glucose analogue (2-[^18^F]fluoro-2-deoxy-D-glucose—[^18^F]FDG) [9]. Other isotopes obtained in the cyclotron that may be used for labeling different substances and using them as radiotracers in PET imaging are carbon-11 (^11^C) with T_1/2_ = 20.4 min, nitrogen-13 (^13^N) − T_1/2_ = 9.96 min, and oxygen-15 (^15^O) − T_1/2_ = 2.03 min. The PET technique also uses isotopes obtained from so-called generators. The two most frequently used generator isotopes in PET are gallium-68 (^68^Ga) − T_1/2_ = 68.3 min and rubidium-82 (^82^Ru) − T_1/2_ = 1.23 min [9].

Methionine, an exogenous essential amino acid with a molar mass of 142.91 g/mol, consisting of a five-carbon chain containing sulfur in its molecule, was first isolated and described at the beginning of the 20th century by John Howard Mueller. Due to its coding by the AUG (adenine-uracil-guanine) codon, which is also the start codon in eukaryotic cells, methionine is the first amino acid residue in all proteins of nuclear organisms. Given that this amino acid is used for the synthesis of almost all proteins and participates in cellular energy production processes, its labeling with radioactive carbon-11 ([^11^C]C-MET) appears to be justified for use in oncological and endocrinological diagnostics, as the increased amino acid metabolism can be detected by this method [10,11,12,13]. During the production of the [11C]C-MET, one of the stable carbon atoms is replaced by an atom of the ^11^C isotope (T_1/2_ = 20.4 min, maximum radiation energy of 960 keV, and maximum tissue range of 4.2 mm), obtained in a cyclotron as a result of bombardment of nitrogen gas or carbon-based precursors by protons (the reaction can be written as ^14^N(p,α)^11^C) [14]. Due to the short half-life, the use of this radiotracer (as well as other ^11^C-based radiotracers) is limited to PET/CT or PET/MRI centers equipped with a cyclotron and a potential patient base that could benefit from this kind of study. Noteworthy, the short half-life of the radiotracer allows for limiting the exposure to ionizing radiation for the patient undergoing the diagnostic procedure, so the radiation dose absorbed by critical organs such as kidneys and liver is low [15,16]. For example, standard [¹¹C]C-MET PET/CT activity provides 4–6 mSv radiation, while 99mTc-Sestamibi SPECT/CT or [¹⁸F]FDG PET/CT gives 8–25 and 7–10 mSv, respectively.

The PET study with the use of the [^11^C]C-MET is conducted about 10–20 min after its intravenous administration, allowing the radiotracer to accumulate and integrate into metabolic pathways. As of the publication date of this article, no serious side effects of the method have been observed. The diagnostic utility of [^11^C]C-MET PET/CT has been demonstrated mainly in oncology diagnostics so far, particularly in neuro-oncology. Studies of myelomas, gliomas, and meningiomas exhibit high sensitivity of the method and radiotracer [17,18,19,20]. Additionally, research undertaken in the field of endocrine tumors provides hope for the potential use of the method in selected patient groups, presenting high sensitivity and diagnostic efficacy rates.

This article delves into the clinical applications and the potential of [^11^C]C-MET PET in the diagnosis of endocrine disorders.

## 2. Methods

Publications available in the PubMed database concerning the use of carbon-11 labeled methionine in PET studies were analyzed. Studies focusing on the use of the marker in oncological diseases affecting non-glandular organs were excluded. The analysis of the remaining 78 articles regarding endocrinological disorders, both prospective and retrospective studies, was performed. The analysis was performed by two researchers who separately reviewed the database, screening abstracts related to the use of the radiotracer, and then analyzed selected articles that met the keywords for full analysis. Finally, 19 studies regarding the use of [^11^C]C-MET PET in the diagnosis of hyperparathyroidism and 10 regarding pituitary adenomas were finally chosen to prepare the article. The methodological path was presented in a chart (Figure 1).

### 2.1. Hyperparathyroidism

Hyperparathyroidism (HPT) is a condition caused by the pathological overproduction of parathyroid hormone (PTH) by the parathyroid glands [21]. There are three types of hyperparathyroidism: primary hyperparathyroidism (pHPT), secondary hyperparathyroidism (sHPT), and tertiary hyperparathyroidism (tHPT) [22,23,24,25]. A characteristic feature of primary and tertiary hyperparathyroidism is hypercalcemia, although normocalcemic forms of pHPT are also recognized [26]. Additionally, patients with pHPT may present with hypercalciuria and hypophosphatemia. The disorders often lead to systemic complications such as nephrolithiasis, soft tissue calcifications, osteodystrophy, osteoporosis, treatment-resistant anemia, and even mental disturbances.

The pHPT results from the presence of a single or multiple parathyroid adenomas (PA), rarely in the course of parathyroid cancer [27]. In sHPT, which is almost only observed in patients with renal function impairment (kidney failure), PTH levels are elevated, mostly due to hyperphosphatemia and deficiency of active vitamin D. In the course of sHPT, hypercalcemia is not observed, which differentiates it from other variants of hyperparathyroidism. However, it has to be mentioned that in some cases of severe intestinal malabsorption, sHPT may develop, even with proper kidney function [28].

Tertiary hyperparathyroidism arises from the autonomization of enlarged parathyroid glands, most commonly in dialysis patients, and is characterized by hypercalcemia and significantly increased PTH concentrations that are often exceeds 1000 pg/mL.

Imaging diagnostics of the parathyroid glands should be practically performed only in order to preoperatively localize the lesions. The primary modalities involve performing neck ultrasound and/or [^99m^Tc]Tc-methoxy-isobutyl-isonitrile ([^99m^Tc]Tc-MIBI) scintigraphy [29,30,31]. Fine-needle aspiration biopsy with PTH washout measurement often provides high diagnostic value in undefined or borderline lesions [32]. The fact that PET/CT imaging is characterized by higher resolution compared to scintigraphy is also reflected in the recommendations on the diagnosis of hyperparathyroidism issued by the European Society of Nuclear Medicine in 2021 [33]. These recommendations indicate the possibility of using the PET/CT method for preoperative localization of the parathyroid glands using radiotracers based on choline derivatives (i.e., carbon-11-labelled choline and fluorine-18-labelled choline) or carbon-11-labelled methionine, especially when other localization techniques of the parathyroid glands fail. Currently, there are no standardized protocols for performing PET/CT examinations in the primary, secondary, or tertiary diagnosis of hyperparathyroidism.

Several studies proved the utility of PET/CT with [^11^C]C-MET usage in cases with negative results of other modalities or in cases of persistent HPT, despite previous surgeries [34]. Below, we present and discuss the current research findings on the use of this radiotracer in the diagnosis of HPT.

### 2.2. Primary HPT

Mathey et al. presented a study on 53 patients with pHPT and negative or inconclusive [^99m^Tc]Tc-MIBI SPECT results, using [^11^C]C-MET and [^18^F]-choline ([^18^F]FCH) PET/CT for diagnosis [35]. PET/CT results were categorized as positive, inconclusive, or negative. The [^18^F]FCH PET/CT was positive in 39 patients (74%), inconclusive in 5 (9%), and negative in 9 (17%). Whereas [^11^C]C-MET PET/CT was positive in 25 patients (47%), inconclusive in 12 (23%), and negative in 16 (30%). Among the 26 patients who underwent surgery, 31 lesions were resected. The [^18^F]FCH PET/CT positively localized 26 lesions in 24 out of 26 patients (92%), whereas [^11^C]C-MET PET/CT localized only 16 lesions in 15 out of 26 patients (58%). The per-patient-based sensitivity and PPV were 96% for [^18^F]FCH and 60% and 94% for [11C]C-MET, respectively (*p* < 0.001). This highlighted the superiority of [^18^F]FCH over [^11^C]C-MET in PA detection.

In the study by Hayakawa on a group of 23 patients with pHPT [36], the study group underwent both [^11^C]C-MET PET/CT and ([^99m^Tc]Tc-MIBI SPECT/CT scanning. Per-patient sensitivities of [^11^C]C-MET PET/CT were 65%, while for MIBI SPECT/CT it was 61%. Per-lesion sensitivities were 91% and 73%, respectively. In conclusion, there was no statistical difference between these two modalities.

In a retrospective analysis by Braeuning et al., 18 patients with pHPT and inconclusive [^99m^Tc]Tc-MIBI SPECT scans underwent PET/CT imaging with [^11^C]C-MET and subsequent surgery (in 12/18 patients) [37]. In 10 patients, single PA was found, while in 2 patients hyperplasia was confirmed histologically. The study correctly localized all adenomas and only 20% of hyperplastic glands. The calculated sensitivity per patient was 91.7%; however, per lesion sensitivity was only 73.3%. [^11^C]C-MET PET/CT scans did not localize lesions smaller than 9 mm (with volume less than 0.2 mL). However, parathyroid adenomas that were found were 5–20 mm, so lesions between 5 and 9 mm seem to be the most difficult to detect. In 6 patients who did not undergo surgery, 5 had a negative or atypical [^11^C]C-MET PET/CT, and further investigation confirmed familial hypocalciuric hypercalcemia (FHH) in 3 of them (true negative studies). In 2 patients, no validation was available, and 1 patient did not agree to surgery.

In a study by Schalin-Jantti, 21 patients with persistent hyperparathyroidism underwent Na [^123^I]/[^99m^Tc]Tc-MIBI SPECT/CT (n = 19), [^11^C]C-MET PET/CT, and selective venous sampling (SVS) (n = 18) prior to reoperation [38]. All patients had undergone one or two previous surgeries. The study revealed localization accuracies of 59% for planar Na[^123^I]/[^99m^Tc]Tc-MIBI, 19% for SPECT/CT, 65% for [^11^C]C-MET PET/CT, and 40% for SVS. In three patients with persistent disease, preoperative Na[^123^I]/[^99m^Tc]Tc-MIBI and SPECT/CT scans were negative, SVS was falsely predictive in all cases, and [^11^C]C-MET PET/CT was negative in only one. Thus, [^11^C]C-MET PET/CT accurately identified the pathological gland in 50% (4/8) of patients with a negative Na[^123^I]/[^99m^Tc]Tc-MIBI scan. Histopathological confirmation of PA was obtained for all these patients.

In a study by Hynh et al., authors conducted a single-center retrospective study involving 45 patients with persistent pHPT [39]. The mean age of the cohort was 57.4 years, with a female predominance (82.2%). Among these patients, 10 had previously unsuccessful surgeries, 12 had PA not localized by other imaging techniques, 16 exhibited discordant results between ultrasonography and [^99m^Tc]Tc-MIBI scintigraphy, and 7 required confirmation of findings from other imaging modalities. The [^11^C]C-MET PET/CT identified potential PA in 32 out of 45 patients (71.1%), with histopathological confirmation in 28 out of 32 patients (62.2%). Within the subgroup of patients with persistent pHPT (ones who underwent previous surgery), the imaging modality identified candidate lesions in 6 out of 10 patients, with histological confirmation in 5 cases. Additionally, [^11^C]C-MET PET/CT successfully localized lesions in 9 out of 12 patients (75.0%) where other imaging modalities failed. Among patients with discordant imaging results prior to [^11^C]C-MET PET/CT, 12 out of 16 (75.0%) demonstrated candidate lesions, of which 10 (62.5%) were confirmed histologically as parathyroid adenomas. In cases where [^11^C]C-MET PET/CT indicated incorrect lesions, successful localization was achieved with subsequent imaging modalities. The study suggested a sensitivity of 70.0% (95% CI: 55.8–84.2%) for [^11^C]C-MET PET/CT in detecting parathyroid adenomas.

Mallikarjuna et al. retrospectively analyzed data from patients with pHPT from 2011 to 2015 [40]. The study included 54 patients (19 males and 35 females). Of the 36 patients who underwent [^99m^Tc]Tc-MIBI scans, 26 were positive. Of the remaining 10 patients, 8 underwent [^11^C]C-MET PET/CT, with 6 positive results. The sensitivity of ultrasonography was 72%, and [^99m^Tc]Tc-MIBI scintigraphy was 70.6%. In patients who were negative on other imaging modalities, the sensitivity of [^11^C]C-MET PET/CT was 71.4%.

Noltes et al. evaluated 32 patients with negative ultrasonography and [^99m^Tc]Tc-MIBI scintigraphy, comparing the diagnostic effectiveness of [^11^C]C-MET PET/CT, [^11^C]-choline PET/CT, and four-dimensional (4D)-CT [41]. The [^11^C]-choline PET/CT was positive in 28 out of 32 patients (88%), [^11^C]C-MET PET/CT was positive in 23 out of 32 (72%), and 4D-CT was positive in only 15 out of 32 patients (47%). Among the 30 patients who underwent surgery, histological confirmation of PA was achieved. The sensitivity of [^11^C]-choline PET/CT, [^11^C]C-MET PET/CT, and 4D-CT was 85%, 67%, and 39%, respectively. Statistical analysis indicated that the sensitivity of [^11^C]-choline PET/CT was significantly higher than that of [^11^C]C-MET PET/CT (*p* = 0.031) and 4D-CT (*p* < 0.001).

A nearly 16-year retrospective study was conducted at a single center to observe patients with primary hyperparathyroidism (pHPT) [42]. Intraoperative localization of the PA was confirmed through fresh frozen section analysis and a significant reduction in intraoperative parathyroid hormone concentration. The study included 658 patients diagnosed with pHPT, excluding 30 patients who underwent surgery for recurrent or persistent disease. The median age of the patients was 58 years (13–39); 71% were female and 29% were male. Neck ultrasound was performed in 576 patients (91.7%), successfully localizing a PA in 481 patients (76.6%). In the remaining 135 patients (23.4%), the preoperative neck ultrasound did not detect a PA. Among this subgroup, [^99m^Tc]Tc-MIBI scintigraphy identified PA in only 34 patients (25.4%). Furthermore, [^11^C]C-MET/[^11^C]-choline PET/CT scans were conducted on this subgroup, accurately identifying PA in 107 patients (79.4%), proving the utility of the method.

Vestergaard et al. performed a prospective cohort study including 27 patients diagnosed with pHPT (18 females, 9 males; mean age 58.9 years) [43]. Among the 33 identified lesion sites, 28 (85%) were histologically verified as PA. The sensitivity and PPV for [^99m^Tc]Tc-MIBI SPECT/CT were 71% and 95%, respectively, while for [^11^C]C-MET PET/CT, they were 82% and 100%. However, the differences were not statistically significant (*p* = 0.38 and *p* = 0.31, respectively).

Iversen et al. conducted a study with 36 pHPT patients, divided into two subgroups: 17 patients underwent [^11^C]C-MET PET/CT before surgery, and 19 underwent [11C]C-MET PET/CT following unsuccessful operations [44]. The [^11^C]C-MET PET/CT identified true-positive pathologic parathyroid glands in 30 out of 36 patients (83%), confirmed histologically. In the pre-surgery subgroup, PET/CT detected parathyroid adenomas in 16 out of 17 cases, while in the post-surgery subgroup, it identified residual or unoperated adenomas in 14 out of 19 cases. The study reported a positive predictive value (PPV) of 91%.

Martínez-Rodríguez et al. conducted a prospective study on 14 patients (mean age 65.5 years) with pHPT [45]. The study compared [^99m^Tc]Tc-MIBI scitigraphy and [^11^C]C-MET PET/CT imaging. Scintigraphy images (planar and SPECT/CT) were obtained 10 min and 2–3 h after the injection of 740 MBq (20 mCi) of [^99m^Tc]Tc-MIBI. PET/CT images were obtained 10 and 40 min after the injection of 740 MBq (20 mCi) of [^11^C]C-MET. The results were positive and concordant in 11 out of 14 patients, correctly localizing PA in 10 out of 11 cases. In three patients, [^99m^Tc]Tc-MIBI scans were positive while [^11^C]C-MET scans were negative; histopathological examinations confirmed PA in two of these three patients.

Hellman et al. in 1994 presented a study on a group of 23 patients with HPT, where PET/CT with [^11^C]C-MET (injected dose 400 to 800 MBq) was performed [46]. Results showed localization of radiotracer in 80% of the patients. Nevertheless, all patients underwent surgery, of whom 13 patients histologically had chief cell adenoma, 10 had parathyroid hyperplasia, and 2 had parathyroid carcinoma. The false negative studies (20%) were small parathyroids in juxtathyroid.

Pogosian et al.’s single-centered retrospective study included 91 patients with pHPT. A total of 91 patients (100%) underwent neck ultrasound, 86 (94.5%) standard CT, 56 (61.5%) [^99m^Tc]-MIBI scintigraphy, and 45 (49.5%) underwent [^11^C]C-MET PET/CT (static scans 10 min after intravenous injection of 350–600 MBq). Imaging studies were followed by minimally invasive parathyroidectomy (PTX) [47]. The sensitivity of [^11^C]C-MET PET/CT was 98%, compared to 75% for CT, 79% for [^99m^Tc]Tc-MIBI scintigraphy, and 67% for ultrasonography. The estimated specificities were 93%, 73%, 75%, and 70%, respectively.

Despite the findings above, the utility of [^11^C]C-MET PET/CT was confirmed by some case studies, such as in a case of a 16-year-old female with negative ultrasonography and [^99m^Tc]Tc-MIBI scintigraphy. The [^11^C]C-MET PET/CT showed focal uptake suggestive of a parathyroid adenoma, which was pathologically confirmed [48]. Additionally, [^11^C]C-MET PET/CT was capable of detecting cases of intrathyroidal parathyroid cancer [49].

### 2.3. Secondary (and Primary) HPT

Rubello et al. performed PET/CT with [^11^C]C-MET in 18 patients (11 women and 7 men) who were hemodialyzed for renal failure (2–14 years’ duration) [50]. Patients had normo-, hypo-, or hypercalcemia; increased PTH concentrations; and parathyroid adenomas were not localized by other nuclear medicine modalities. In 3/10 of patients with normo- or hypocalcemia, increased [^11^C]C-MET accumulation was diagnosed in one gland. In 7/8 patients with hypercalcemic HPT, studies showed increased uptake in one or more glands.

Sundin et al. presented data on 34 patients with pHPT (n = 32) and sHPT (n = 2) [51]. These patients underwent [^11^C]C-MET PET/CT before either primary (n = 9) or re-operative (n = 25) surgery. Lesion localization was confirmed in 29 out of 34 patients (85%), with no false-positive results, indicating a specificity of 100%. Localization success rates were 78% for primary tumors and 88% for re-operative cases. Corresponding accuracy rates were 59% for CT and 55% for ultrasound in primary cases, and 57% for CT and 52% for ultrasound in re-operative cases.

Tang et al. studied 30 patients, including 22 with pHPT and 8 with sHPT [52]. The sensitivity of [^11^C]C-MET PET/CT was 92% compared to 95% for [^99m^Tc]Tc-MIBI scintigraphy in detecting adenomas and 68% versus 59% for detecting hyperplasia.

Otto et al. conducted a study on 30 patients with hyperparathyroidism (HPT), including 16 with pHPT, 12 with sHPT, and 2 with parathyroid carcinoma [53]. In the group of 12 patients with sHPT, postoperative histopathology identified 36 parathyroid glands. Of these, 25 were visible on [^11^C]C-MET PET/CT compared only to 17 on [^99m^Tc]Tc-MIBI scintigraphy. In the group with pHPT and carcinoma, [^11^C]C-MET PET/CT detected lesions in 17 out of 18 cases, compared to 9 out of 18 cases with [^99m^Tc]Tc-MIBI scintigraphy.

### 2.4. Tertiary (and Primary) HPT

In a study by Lenschow et al., 14 patients with pHPT and 3 patients with tHPT underwent neck ultrasound (USG) and [^99m^Tc]Tc-MIBI SPECT [54]. [^11^C]C-MET PET/CT was carried out only in patients with negative [^99m^Tc]Tc-MIBI results. While USG localized a parathyroid adenoma in 10/17 patients (59 %) cases, a [^99m^Tc]Tc-MIBISPECT/CT identified them in 11/17 (65 %) cases. In remaining patients, [^11^C]C-MET PET/CT found adenomas in 5/6 cases.

Cook et al. in a study on 5 patients with pHPT and 3 with tHPT who had undergone previous surgery showed that [^11^C]C-MET PET/CT scans showed radiotracer accumulation in 5/5 patients with pHPT and 1/3 with tHPT [55].

Kołodziej et al. examined 19 patients with tertiary hyperparathyroidism (tHPT) resistant or intolerant to non-invasive treatment, who had negative scintigraphy and neck ultrasonography results [56]. Positive [^11^C]C-MET PET/CT results were obtained in 17 out of 19 patients. Nine of these 17 patients underwent surgery, and all were confirmed histologically as having parathyroid adenomas, demonstrating a sensitivity of 100%.

### 2.5. Pituitary Tumors

Pituitary adenomas are a type of benign tumor originating from the glandular part of the pituitary gland [57]. They belong to a specific group of neuroendocrine tumors called PitNENs [58]. They can be divided into hormonally active and inactive types. Hormonal adenomas may secrete growth hormone (GH), causing acromegaly; thyroid-stimulating hormone (TSH), causing secondary hyperthyroidism; and adrenocorticotropic hormone (ACTH), being the most common cause of Cushing’s syndrome (CS) [59,60,61,62]. Adenomas may also secrete prolactin (PRL), causing hyperprolactinemia [63]. Tumors secreting follicular stimulating hormone (FSH) and luteinizing hormone (LH) can also present with symptoms such as libido disorders or menstrual irregularities. However, adenomas that secrete the alpha-subunit of glycoprotein hormones (a-SU) or those that do not secrete any tropic hormones are classified as “inactive” adenomas [64,65,66]. Larger tumors, in addition to their hormonal activity, may cause the tumor mass effects causing headaches and vision impairment (mostly limiting marginal vision by compression of the optic chiasm). The test of choice in the diagnosis of pituitary adenomas is still MRI; however, the smallest adenomas remain often undetectable with standard diagnostic methods, posing the greatest diagnostic challenges and requiring higher-reference testing [67,68].

### 2.6. Non-Functioning Tumors and Mixed Groups

Over 40 years ago, Bergström et al. conducted a study involving four patients diagnosed with hormonally inactive pituitary adenomas using PET/CT with L-[methyl-^11^C]methionine and D-[methyl-^11^C]methionine injections [69]. Enantiomer L[^11^C]C-MET emerged as a promising modality for the detection of pituitary adenomas.

Tomura et al. investigated 77 patients in a subsequent study utilizing PET/CT with [^11^C]C-MET [70]. Maximum standardized uptake value (SUV_max_) measurements were taken for the pituitary gland in healthy individuals, revealing an average SUV_max_ of 2.60 ± 1.04. Tomura et al. also demonstrated a negative correlation between SUV_max_ and patient age through linear regression analysis. Furthermore, there was no significant difference in SUV_max_ between male and female subjects. These findings substantiate the strong accumulation of [^11^C]C-MET in the normal pituitary gland, thereby proposing its utility in distinguishing between normal and abnormal pituitary tissues.

In another investigation by Tang et al., 33 patients who had undergone pituitary surgery—24 with hormone-secreting adenomas and 9 with non-functional adenomas—underwent postoperative evaluation with [^11^C]C-MET PET/CT due to residual tissue detected in MRI or persistent biochemical tumor activity [71]. Among these patients, [^11^C]C-MET PET/CT detected pathologically hypermetabolic tissue in 30 out of 33 cases. In 14 of these 30 cases, MRI failed to differentiate between residual adenoma and postoperative scarring. Both positive results from [^11^C]C-MET PET/CT and MRI were confirmed in only 16 patients. Notably, there was only one instance where neither MRI nor [^11^C]C-MET PET/CT detected adenomatous tissue. The study successfully located all non-functional adenomas using [^11^C]C-MET PET/CT. Furthermore, while MRI identified only 1 out of 8 adrenocorticotropic hormone-secreting adenomas, [^11^C]C-MET PET/CT detected all 8 cases.

### 2.7. Prolactinomas (PRL-OMAS)

Bashari et al. investigated 13 patients with pituitary microprolactinomas, among whom 11 were intolerant and 2 were resistant to dopamine agonist therapy [72]. In all cases, [^11^C]C-MET PET/MR demonstrated focal tracer uptake. Localization within the sella turcica (depression in sphenoid bone where pituitary is located) was confirmed in 12 cases, with residual tissue identified in the cavernous sinus in one patient who had undergone prior surgery. This patient subsequently underwent PET-guided stereotactic radiosurgery, resulting in nearly complete normalization of serum prolactin levels. Five patients underwent endoscopic transsphenoidal selective adenomectomy, achieving complete remission of hyperprolactinemia and normalization of other pituitary functions. The remaining patients either awaited treatment, declined further invasive procedures, or did not qualify for such interventions.

### 2.8. Somatotropinomas (GH-Omas)

Haberbosch et al. examined 189 cases of acromegaly over a 12-year period at a single reference center [73]. Among these cases, 61 patients underwent [^11^C]C-MET PET/CT due to inconclusive results from MR (n = 38, 62.3%), residual tumor detection (n = 14, 23.0%), surgical planning (n = 6, 9.8%), or identification of *de novo* tumors (n = 3, 4.9%), and results showed a potential number of 52 locations. Ultimately, 33 out of 61 patients (54.1%) proceeded to undergo surgery. Of these, 24 patients (72.7%) achieved complete biochemical remission postoperatively. In the remaining 9 cases, insulin-like growth factor 1 (IGF-1) concentration was reduced to less than twice the upper limit of normal (ULN), with 6 out of these 9 cases achieving a concentration less than 1.1 times the ULN.

Koulouri et al. investigated 26 patients with persistent acromegaly following previous treatments, where MRI findings were inconclusive [74]. Each patient underwent [^11^C]C-MET PET/MRI, revealing tracer uptake solely within the normal pituitary gland in the 4 patients who achieved complete remission after initial surgery. Conversely, in the 26 patients with active acromegaly, [^11^C]C-MET PET/MRI identified abnormal tracer uptake in 25 out of 26 cases. Based on these findings, 14 subjects underwent endoscopic transsphenoidal surgery, resulting in significant improvement (n = 7) or complete resolution (n = 7) of residual acromegaly. One patient received stereotactic radiosurgery, and two patients with cavernous sinus invasion underwent image-guided fractionated radiotherapy, achieving effective disease control. Three subjects are awaiting further intervention, while five patients opted for adjunctive medical therapy.

Rodriguez-Barcelo et al. conducted a prospective observational study that included 17 patients, of whom 6 (2 microadenomas and 4 macroadenomas) had a new diagnosis of acromegaly and 11 had previously undergone resection of a somatotropinoma [75]. Modality demonstrated that in the majority of cases, increased uptake is coincident with the location of the pituitary lesion or tumor remnants in MRI. In the postoperative group, the evidence of residual tumor was also confirmed in all patients with active disease (n = 7). Sensitivity and specificity of the method were 86% for the diagnosis of recurrence.

### 2.9. Thyrotropinomas (TSH-Omas)

Gillet et al. presented a study involving 6 healthy volunteers and 10 patients with thyrotropinoma confirmed by elevated free thyroid hormone concentration and lack of TSH suppression [76]. The study group consisted of 2 men and 8 women, with a mean age of 58 years (range 27–75). In the study, the presence of pituitary adenomas was confirmed in PET/CT with [^11^C]C-MET. The median activity administered was 382 MBq (range 293–411 MBq). Authors proved that hormonal suppression with the use of lanreotide and the use of normalized subtraction imaging before and after hormonal suppression of the tumor can improve PET/CT accuracy. [^11^C]C-MET scans can be normalized to the cerebellum to minimize the effects of physiological variation in [^11^C]C-MET uptake in the pituitary and to improve detection of lateralized adenomas.

### 2.10. Corticotropinomas (ACTH-Omas—Cushing Disease)

Ishida et al. conducted a retrospective analysis of 15 patients with recurrent CD to evaluate the role of [^11^C]C-MET PET/CT in distinguishing equivocal MRI findings between recurrent tumors and postsurgical cavities, thereby guiding further treatment decisions [77]. All patients had undergone transsphenoidal surgery (TSS), with most having undergone multiple surgeries and confirmed corticotrope tumors with hypercortisolemia. MRI scans showed less-enhanced lesions that were challenging to differentiate from postsurgical changes. [^11^C]C-MET uptake was positive in 8 patients (9 examinations) and negative in 7. Corticotrope tumors were identified in all 5 patients, despite one having negative [^11^C]C-MET uptake. Notably, [^11^C]C-MET uptake precisely identified tumor locations contrary to those suspected by MRI in 2 patients. Interestingly, [^11^C]C-MET uptake disappeared in one patient after temozolomide treatment.

Berkmann et al. conducted a retrospective double-center cohort study involving 15 patients who underwent transsphenoidal surgery for biochemically proven CD [78]. Six patients received either [^11^C]C-MET PET/MRI or (18-fluoro-ethyl-L-thyrozine) [^18^F]FET PET/MRI, and three patients underwent both examinations. MRI detected 67% of the tumors ([^11^C]C-MET group 56%, [^18^F]FET group 78%). All tumors were microadenomas. [^18^F]FET PET/MRI results correlated positively with intraoperative and histopathological tumor localization in all cases, achieving a sensitivity and specificity of 100% (95% CI 66.37–100%). One [^11^C]C-MET PET/MR indicated contralateral localization compared to the expected site, resulting in a sensitivity and specificity of [^11^C]C-MET PET/MRI for tumor localization of 89% (95% CI 51.7–99.7%).

Koulouri et al. conducted a study involving 20 patients: 10 with de novo Cushing’s disease, 8 with residual or recurrent hypercortisolism following initial pituitary surgery and/or radiotherapy, and 2 with ectopic Cushing’s syndrome [79]. All patients underwent [^11^C]C-MET PET/CT. In 7 out of 10 patients with de novo CD, asymmetric [^11^C]C-MET uptake was observed within the pituitary gland, corresponding with suspected corticotrope microadenomas visualized on MRI. Histological confirmation was subsequently obtained in all cases following transsphenoidal surgery. Focal [^11^C]C-MET uptake correlating with suspected abnormalities on pituitary MRI was seen in 5 out of 8 patients with residual or recurrent CS. In 2 patients with ectopic CS, [^11^C]C-MET uptake was concentrated in distant metastatic sites with minimal uptake in the sellar region, confirming neoplastic foci.

Feng et al. studied 43 patients with pituitary disorders: 15 with Cushing’s disease, 16 with acromegaly, and 12 with prolactinoma [80]. All patients underwent [^18^F]FDG PET/CT, and 39 of them underwent [^11^C]C-MET PET/CT. On [^18^F]FDG PET/CT, 29 out of 43 patients (67.0%) had positive findings, all of which were confirmed true positives. On [^11^C]C-MET PET/CT, 37 out of 39 patients (95%) had positive results, with only 2 cases showing false positives. Surgical intervention was performed in all cases, and among the 12 patients with positive [^11^C]C-MET PET/CT findings, [^18^F]FDG PET/CT results were negative. An interesting fact is that in the study, all patients with CD had positive results of the [^11^C]C-MET PET/CT study.

Ikeda et al. conducted a study involving 35 patients diagnosed with Cushing’s disease confirmed by surgical pituitary exploration [81]. The cohort included 20 cases of overt CD and 15 cases of preclinical CD. Utilizing superconductive MRI (1.5 or 3.0 T) and composite images from [^18^F]FDG PET or [^11^C]C-MET PET combined with 3.0-T MRI, the localization of adenomas was compared to surgical findings. Superconductive MRI demonstrated a diagnostic accuracy of only 40% in identifying the location of CD adenomas. This was attributed to false-negative results in 10 cases, false-positive results in 6 cases, and instances of double pituitary adenomas in 3 cases. In contrast, the accuracy of microadenoma localization using [^11^C]C-MET PET/MRI was 100%, while that of [^18^F]FDG PET/MRI was 73%. There was no significant difference in the SUV_max_ of adenomas evaluated by [^11^C]C-MET PET between preclinical and overt CD cases.

Moreover, there is a presence of case studies proving the utility of the method in single patients with challenging patients. Lurquin et al. presented a case study of a 48-year-old woman with a high clinical suspicion of ACTH-dependent Cushing’s syndrome [82]. Pituitary MRI did not reveal any adenoma, but bilateral inferior petrosal sinus sampling (BIPSS) confirmed central ACTH secretion and revealed a significant right-to-left gradient. This prompted exploratory hypophysectomy, which did not lead to a cure. Subsequent [^11^C]C-MET PET/MRI identified a hypermetabolic lesion adjacent to the posterior wall of the sphenoidal sinus. Surgical resection confirmed this lesion as an ectopic ACTH-secreting pituitary adenoma expressing ACTH and T-Pit, resulting in complete remission of hypercortisolism.

### 2.11. Pitfalls in Diagnostics

Despite the high sensitivity and accuracy of [^11^C]C-MET PET, it must be remembered that there might be false-positive study results leading to unnecessary surgical interventions. Adler et al. presented a case of a false-positive parathyroid scan that convinced surgeons to perform sternotomy. The detected lesion appeared to be a thymoma, which in the majority of cases is benign and harmless [83]. Mahajan et al., during the diagnostic process of pHPT, also located a lesion in the thyroid, which after thyroidectomy appeared to be only a [^11^C]C-MET PET positive colloid nodule [84]. In a retrospective study from 2016, Noltes et al. reported that in one of the patients with pHPT, histopathological examination revealed papillary thyroid carcinoma in a focal lesion accumulating a [^11^C]C-MET assessed in PET/CT as a parathyroid gland [85].

## 3. Discussion

The results of presented studies underscore the clinical utility of [^11^C]C-MET PET in the precise localization and evaluation of pituitary adenomas, particularly in challenging diagnostic scenarios and postoperative follow-up. However, there are still relatively low numbers of hard evidence and data regarding standardized activity of radiotracer, and optimal time of its application and start of acquisition, and detailed study protocol. Despite being promising and potential, the method seems to require more reliable studies. Nevertheless, summarizing the studies regarding HPT, [^11^C]C-MET PET/CT was able to localize parathyroid adenoma or other causes of HPT in a total of 448 over 595 (75.3%). In the studies comparing [^11^C]C-MET PET/CT directly (“head to head”) with other modalities (like USG and [^99m^Tc]Tc-MIBI scintigraphy), the positive results were 276/356 (77.5%).

In the studies regarding pituitary adenomas, [^11^C]C-MET PET added diagnostic value in 241/274 (88.0%) cases. Most interestingly, in cases of CD, the positive results were obtained in 107/124 (86.3%) patients, which seems to be a relatively high number, knowing the potential difficulties of the diagnostic process of CD and the fact that presently the gold standard of diagnosis is invasive bilateral inferior petrosal sinus sampling (BIPSS) [86].

The differences in results of the presented studies may result from the heterogeneity of the study groups. However, the sensitivity and accuracy of the study still should be considered as high compared to most commonly used methods like USG and scintigraphy for HPT and MRI for pituitary adenomas. Whereas up to date meta-analysis regarding the utility of [^99m^Tc]Tc-MIBI presented various results (sensitivity from 58 to 84%), the [^11^C]C-MET PET/CT seems to be more accurate and provides better scan resolution due to the characteristics of the PET itself [87,88]. Particular attention should also be paid to the fact that most studies did not take into account parameters such as a similar radioisotope activity, use of different scanners (gamma cameras), or the use of drugs that may influence the results, such as calcium channel blockers [89]. Moreover, in some of the studies, the [^11^C]C-MET PET/CT was performed only if [^99m^Tc]Tc-MIBI scintigraphy failed in the localization of the lesions; thus, the study can be recommended only for patients who were not diagnosed with the use of standard methods and procedures during the qualification for the surgery.

The potential limitations of the [^11^C]C-MET are: cost of the radiotracer resulting from non-standard utilization of cyclotron radioisotope production, low tumor specificity, as methionine can be used by other types of tumors, and above all, problematic study organization resulting from the short half-life of [^11^C]C-MET and the necessity of “having” a patient at the PET/CT site even before the production of the radioisotope is started.

Due to our observations, patients who could benefit most from [¹¹C]C-MET studies are patients in which other modalities failed to detect parathyroid or pituitary adenomas or were inconclusive. Especially ones with potentially ectopic lesions, multiple neoplasms (MEN syndrome, or polyglandular disease), or patients who underwent ineffective surgery.

Direct comparison of the [^11^C]C-MET study to the [¹⁸F]FDG study is difficult as parathyroid lesions due to their relatively low metabolism and glucose usage are difficult. However, [^11^C]C-MET predominates for the diagnosis of pituitary and parathyroid adenomas due to aforementioned differences in tumor glucose metabolism (radiotracer has higher specificity for amino acid metabolism), high brain glucose uptake, which can limit [¹⁸F]FDG sensitivity and specificity, and lower radiation exposure resulting from the over 5x shorter half-life of [^11^C]C-MET.

In the diagnosis of pituitary adenomas, the [^11^C]C-MET PET seems to be a very reliable method for detecting not only the morphological presence of neoplasms but also their amino-acid metabolism, helping in qualification for the surgery or re-surgery. The use of the carbon-11 radiotracer and the PET gives the opportunity to detect lesions that are not able to be detected in standard procedures. However, the dearth of prospective studies on a large group of patients may nowadays limit the wide use of the method. Additionally, technical issues and the short half-life of carbon-11 may add another limitation for researchers in projecting high-value studies. Nevertheless, we consider [^11^C]C-MET PET/CT or MRI as a method, which can bring significant results leading to a change in the diagnostic and therapeutic process. Moreover, rapid development of nuclear medicine could encourage researchers to develop population multi-center studies for method validation, which can provide standardization of procedures and individualization of radiotracer activity used for the study, as well as decrease the cost of the radioisotope production and study itself.

## 4. Conclusions

Studies using carbon-11 labeled methionine appear to be a significant addition to the diagnostic workup of tumors originating from both the parathyroid glands and pituitary gland. In the studies presented above, the use of [^11^C]C-MET PET has demonstrated its utility, particularly in cases that were undiagnosable using standard methods, showing its high sensitivity. It is important to note that due to technical aspects such as the half-life of carbon-11 and challenges in its production, these studies are intended for a highly selected group of patients for whom other diagnostic methods have indeed failed or are not feasible and where clinical symptoms resulting from the presence of hormonally active lesions could lead to significant health deterioration.

## Figures and Tables

**Figure 1 pharmaceuticals-18-00229-f001:**
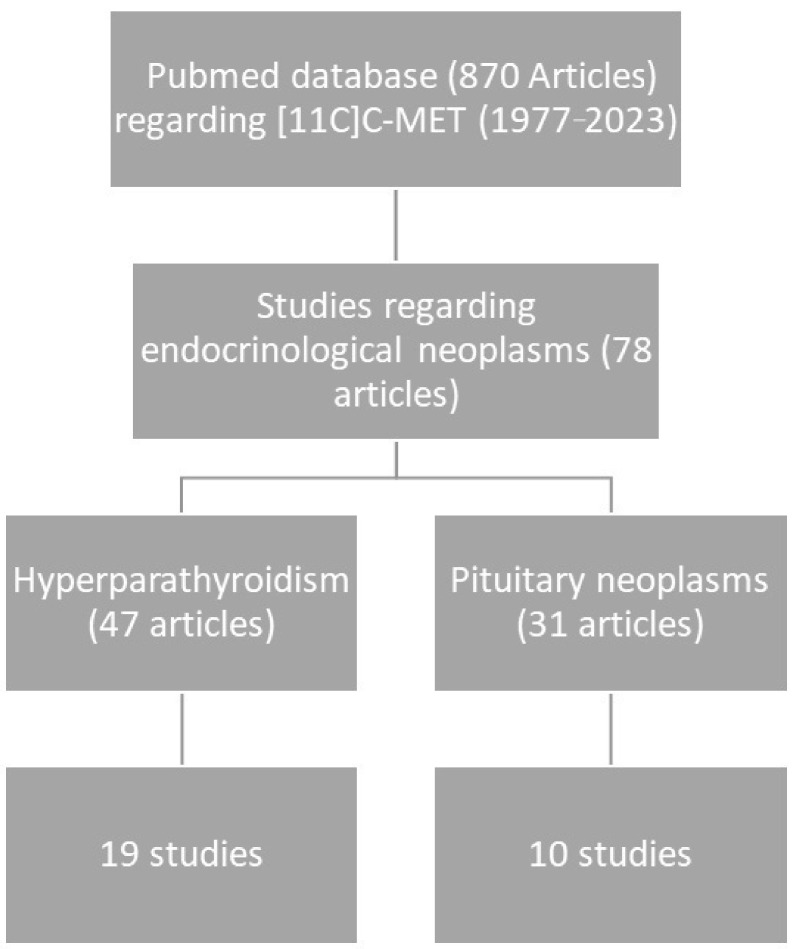
A chart presenting the process of articles selection for preparing the review.

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
