# Peer review of "The Use of [11C]C-Methionine in Diagnostics of Endocrine Disorders with Focus on Pituitary and Parathyroid Glands"

_pharmaceuticals, 2025, doi:10.3390/ph18020229_

Round 1
Reviewer 1 Report
Comments and Suggestions for Authors
The content of the manuscript submitted for assessment is a retrospective study dealing with the use of 11C-methionine in the diagnosis of hyperparathyroidism and pituitary neoplasia. In total, around 30 original papers were analyzed. The submitted manuscript tries to evaluate the position of PET diagnostics using methionine in comparison with fluorinated analogs (FET) and SPECT radiopharmaceuticals, especially MIBI.
Such a comparison is difficult to understand from the point of view of a nuclear chemist and a radiopharmacist due to the different pharmacokinetics and dynamics of the compared radiopharmaceuticals. Apart from the comparison of different radiopharmaceuticals, differing in the radionuclides used, the physico-chemical aspect plays an essential role - i.e. the radioactive transformation itself and the emission of radiation with different energy in the case of C-11 and F-18, and in the case of 99mTc also a completely different nature of nuclear transformation. However, the clinic considers more information how the compared radiopharmaceuticals contribute to a timely and accurate diagnosis.
I am convinced that the manuscript will find its readers among clinical workers focusing on nuclear endocrinology.
The work is methodically and linguistically well written. Unfortunately, I have to state that the introduction is too boring, because from line 24 almost to 57 the basic principles are described, which it is not necessary to repeat here among the professional public and emphasize the principles of PET and SPECT in such detail. And I would absolutely recommend avoiding the description of FDG, which is related to the text of the study only by the fact that it is a radiopharmaceutical for PET. [11C]methionine itself is a substance already so notorious that it does not need too much explanation.
However, there are a number of errors in the work related to the correct writing of [11C] methionine and the index of radionuclides in general, e.g. lines 66, 68, 151, 183, 204, 229, 255,331. Therefore, I recommend paying particular attention to checking typographical errors.
Some paragraphs in the text are in a different font size, line 31 - 43
line 495 should be... scintigraphy instead of scinitigraphy
I have two fundamental questions about the work:
1) Can you please explain to me how the irradiation of nitrogen disks with protons is meant? As a radiochemist, I have to strongly disagree with such a statement. The construction of the target is different here, and the statement you make seems misleading to me.
2) Can you please explain what is meant by "unified radioisotope activity"? Do you mean unified applied activity for a specific radiopharmaceutical? However, the radiological standards (guidelines) take this into account. In this regard, it is no longer possible to compare PET and SPECT. Can you explain this statement?
I evaluate the manuscript as interesting and summarizing the findings of a number of studies. I recommend it for considering after performing minor revisions.
Author Response
Dear Reviewer,
First of all, we would kindly like to thank you for the review. We have corrected the manuscript according to all your comments and suggestions. We hope the corrected manuscript will meet your all expectations. Below, we have attached the answers for all of your questions.
Such a comparison is difficult to understand from the point of view of a nuclear chemist and a radiopharmacist due to the different pharmacokinetics and dynamics of the compared radiopharmaceuticals. Apart from the comparison of different radiopharmaceuticals, differing in the radionuclides used, the physico-chemical aspect plays an essential role - i.e. the radioactive transformation itself and the emission of radiation with different energy in the case of C-11 and F-18, and in the case of 99mTc also a completely different nature of nuclear transformation. However, the clinic considers more information how the compared radiopharmaceuticals contribute to a timely and accurate diagnosis. I am convinced that the manuscript will find its readers among clinical workers focusing on nuclear endocrinology.
We completely agree that from the point of view of a nuclear chemist or radiopharmacist comparing different radiotracer is inappropriate, however the aim of our article was to present clinical aspect and capabilities of the studies using 11C-MET.
The work is methodically and linguistically well written. Unfortunately, I have to state that the introduction is too boring, because from line 24 almost to 57 the basic principles are described, which it is not necessary to repeat here among the professional public and emphasize the principles of PET and SPECT in such detail. And I would absolutely recommend avoiding the description of FDG, which is related to the text of the study only by the fact that it is a radiopharmaceutical for PET. [11C]methionine itself is a substance already so notorious that it does not need too much explanation.
We have updated introduction.
However, there are a number of errors in the work related to the correct writing of [11C] methionine and the index of radionuclides in general, e.g. lines 66, 68, 151, 183, 204, 229, 255,331. Therefore, I recommend paying particular attention to checking typographical errors.
We have checked and updated the detectable typos and errors. We tried to keep the correct nomenclature of radioisotopes. Superscript were updated.
https://www.sciencedirect.com/science/article/pii/S0969805117303189?via%3Dihub
Some paragraphs in the text are in a different font size, line 31 – 43
We have updated the front size.
line 495 should be... scintigraphy instead of scinitigraphy
We have corrected the text.
- Can you please explain to me how the irradiation of nitrogen disks with protons is meant? As a radiochemist, I have to strongly disagree with such a statement. The construction of the target is different here, and the statement you make seems misleading to me.
We have corrected the text. The unclear description resulted from a and a clinical origin of authors (we are all physicians) and translating mistake.
- Can you please explain what is meant by "unified radioisotope activity"? Do you mean unified applied activity for a specific radiopharmaceutical? However, the radiological standards (guidelines) take this into account. In this regard, it is no longer possible to compare PET and SPECT. Can you explain this statement?
The “unified radioisotope activity” was referring to similar activity “value”. In majority of the studies activities 400-600 MBq were used. We have changed unified to similar.
Reviewer 2 Report
Comments and Suggestions for Authors
The study titled "The use of [11C]C-methionine in diagnostics of endocrine disorders with focus on pituitary and parathyroid adenomas" is very interesting. It contains comprehensive review of an emerging nuclear medicine technique for diagnosing endocrine disorders, detailed exploration of [11C]C-methionine PET/CT in hyperparathyroidism and pituitary adenomas. However, I have some questions could be answered and added to the article:-
1. What are the main advantages and limitations of using [11C]C-methionine as a radiotracer in PET imaging for endocrine disorders?
2. How does the sensitivity and specificity of [11C]C-methionine PET/CT compared to other established imaging modalities for detecting parathyroid adenomas?
3. What is the minimum size of parathyroid lesions that can be reliably detected using [11C]C-methionine PET/CT, based on the findings from Braeuning et al. [37]?
4. Are there any specific patient populations or clinical scenarios where [11C]C-methionine PET/CT might be particularly useful or recommended over other imaging modalities?
5. How does the radiation exposure from [11C]C-methionine PET/CT compare to other diagnostic modalities used for parathyroid imaging?
6. What further research is needed to establish [11C]C-methionine PET/CT as a standard diagnostic tool for parathyroid disorders?
Finally, the manuscript presents a well-structured, scientifically sound review that contributes meaningful knowledge to the field of nuclear medicine and endocrine diagnostics, making it worthy of publication.
Author Response
Dear Reviewer,
First of all, we would kindly like to thank you for the review. We have updated the manuscript according to all your comments and suggestions. We hope the corrected manuscript will meet your all expectations. Below, we have attached the answers for all of your questions.
- What are the main advantages and limitations of using [11C]C-methionine as a radiotracer in PET imaging for endocrine disorders?
Paragraph regarding limitations was added.
- How does the sensitivity and specificity of [11C]C-methionine PET/CT compared to other established imaging modalities for detecting parathyroid adenomas?
Paragraph regarding comparing studies was added.
- What is the minimum size of parathyroid lesions that can be reliably detected using [11C]C-methionine PET/CT, based on the findings from Braeuning et al. [37]?
Paragraph was updated.
- Are there any specific patient populations or clinical scenarios where [11C]C-methionine PET/CT might be particularly useful or recommended over other imaging modalities?
Paragraph was updated.
- How does the radiation exposure from [11C]C-methionine PET/CT compare to other diagnostic modalities used for parathyroid imaging?
Paragraph was updated.
- What further research is needed to establish [11C]C-methionine PET/CT as a standard diagnostic tool for parathyroid disorders?
Paragraph was updated.
Reviewer 3 Report
Comments and Suggestions for Authors
The manuscript, titled “The use of [11C]C-methionine in diagnostics of endocrine disorders with focus on pituitary and parathyroid adenomas,” provides a comprehensive analysis of the diagnostic utility of [¹¹C]C-Methionine in various malignancies, as reported in 78 studies published between 1977 and 2023. The review primarily focuses on its application in hyperparathyroidism (19 studies) and pituitary adenomas (10 studies). Despite the challenges associated with [¹¹C]C-Methionine, including its physical properties and the complexities of its production process, existing research highlights its effectiveness as a PET radiotracer. It has proven valuable in detecting lesions that are often difficult to identify using other imaging modalities.
Although this manuscript is well-written and well-organized, it would benefit if authors could highlight the major advantages of [¹¹C]C-Methionine PET imaging compared to [¹⁸F]-labeled tracers.
Author Response
Dear Reviewer,
First of all, we would kindly like to thank you for the review. We updated the discussion according to your suggestion. We hope the corrected manuscript will meet your expectations.
Although this manuscript is well-written and well-organized, it would benefit if authors could highlight the major advantages of [¹¹C]C-Methionine PET imaging compared to [¹⁸F]-labeled tracers.
Discussion was updated, and paragraph regarding [¹⁸F]-labeled radiotracers added.